# Cerebral Neural Changes in Venous–Arterial Extracorporeal Membrane Oxygenation Survivors

**DOI:** 10.3390/brainsci13040630

**Published:** 2023-04-06

**Authors:** Jueyue Yan, Zhipeng Xu, Xing Fang, Jingyu You, Jianhua Niu, Mi Xu, Jingchen Zhang, Jia Hu, Xujian He, Tong Li

**Affiliations:** Department of Critical Care Medicine, The First Affiliated Hospital, Zhejiang University School of Medicine, No. 79 Qingchun Road, Hangzhou 310003, China

**Keywords:** extracorporeal membrane oxygenation, cognition, magnetic resonance imaging, degree centrality

## Abstract

Background: Extracorporeal membrane oxygenation (ECMO) is used as temporary cardiorespiratory support in patients with critical ailments, but very little is known about the functional cerebral changes in ECMO survivors. Degree centrality (DC), a graph-based assessment of network organization, was performed to explore the neural connectivity changes in ECMO survivors compared to controls and their correlation with cognitive and neurological measures. Methods: This exploratory observational study was conducted from August 2020 to May 2022. ECMO survivors and controls underwent functional magnetic resonance imaging (fMRI) of the brain. We performed DC analysis to identify voxels that showed changes in whole-brain functional connectivity with other voxels. DC was measured by the fMRI graph method and comparisons between the two groups were performed. All participants underwent neuropsychological assessment (Montreal Cognitive Assessment, MoCA). Blood serum neuron-specific enolase and the Glasgow Coma Scale (GCS) were assessed in ECMO survivors. Results: DC values in the right insula and right precuneus gyrus were lower in ECMO survivors and higher in the right medial superior frontal gyrus compared to controls (all *p* < 0.001). Decreased connectivity in the right insular and right precuneus gyrus correlated with total MoCA scores, delayed recollection, and calculation (all *p* < 0.05). Increased serum NSE levels, GCS score, and GCS–motor response correlated with decreased connectivity in the right insular and right precuneus gyrus and increased connectivity in the right medial superior frontal gyrus (all *p* < 0.05). Conclusions: We showed that both functional impairment and adaptation were observed in survivors of ECMO, suggesting that neural connectivity changes may provide insights into the mechanisms that may potentially link ECMO survivors to neurological and cognitive disorders.

## 1. Background

Cardiogenic shock (CS) is the most severe and life-threatening condition associated with acute heart failure (HF) and is characterized by end-organ hypoperfusion resulting from the low cardiac output. Extracorporeal life support (ECLS), especially venous–artery extracorporeal membrane oxygenation (VA-ECMO), is widely used in patients presenting with refractory cardiogenic shock. Refractory cardiogenic shock is an indication of VA-ECMO. In critically ill patients who require cardiac and/or pulmonary support as a bridge to recovery, extracorporeal membrane oxygenation (ECMO) provides temporary mechanical circulatory support [1,2]. According to the Extracorporeal Life Support Organization (ELSO), the use of ECMO has increased worldwide and can be performed in different settings when suggested [1]. Although ECMO is suggested to be beneficial to patients, it has been suggested that ECMO is linked with a high rate of complications [3]; the patients’ underlying clinical manifestations and phase of illness may contribute to the progression of these complications.

Neurologic complications occur during or immediately after ECMO and are associated with poor survival [4,5,6]. Acute ischemic stroke, intracranial hemorrhage, seizures, cognitive decline, and brain death are a few of the short-term complications [7]. The occurrence and death of neurological complications differ across different ECMO centers in adults. However, neurological complications may still occur after ECMO, and these complications are too often overlooked; some patients with neurological complications may have been diagnosed with organ failure, and thus may have died without undergoing neuroimaging, while on other occasions, insufficient routine neurological monitoring and standardized diagnostic criteria may have masked the prevalence of neurologic complications [7,8].

To date, reports on cerebral changes in ECMO survivors are very limited. Neurologic injury reports have been largely reliant on neuroimaging and electroencephalogram (EEG) performed in the setting of clinical suspicion. Clinical neurologic evaluation is often confused by sedation or the neuromuscular blockade required to prevent the dislodgement of ECMO cannulas, and neuroimaging is restricted to computed tomography (CT) because the ECMO circuit is incompatible with MRI. This can leave patients with obscured neurologic injury that may occur before ECMO due to hemodynamic variability and hypoxia or during ECMO from a host of processes such as thromboembolism, hemodynamic changes, and endothelial inflammation.

Our current study aimed to explore the intrinsic dysconnectivity pattern in the whole brain functional network at the voxel level in ECMO survivors compared to controls. We also explored the DC changes in ECMO survivors and their association with their cognition and their neurological assessment.

## 2. Methods

This exploratory observational study was conducted at the Intensive Care Unit Department of First Affiliated Hospital of Zhejiang University, School of Medicine from August 2020 to May 2022. The protocols of the study were approved by the Ethics Committee of the First Affiliated Hospital of Zhejiang University (No. 2020-IIT-1163) and followed the tenets of the Declaration of Helsinki. Written informed consent was obtained from each participant or legal guardian of the participant before enrollment in our study. 

Our inclusion criteria for ECMO were as follows: (1) Previously independent in daily living activities. (2) No known underlying neurological disorders. (3) Peripheral intubation was performed for a cardiogenic shock with VA ECMO support. (4) The patient was over 18 years old. Of the 188 patients were treated with ECMO at our center, 144 underwent VA, while 44 underwent VV. In the cases included in the analysis, these were due to severe circulatory failure, requiring high-dose vasoactive drugs to maintain organ perfusion for VA ECMO support. Of the 144 VA ECMO patients included, those supported by ECMO due to non-cardiogenic factors were excluded. All study cases were performed by peripheral single-lumen intubation. The selected vessels were all femoral arteries and femoral veins. Cannulas were placed percutaneously by vessel puncture, guidewire placement, and serial dilation (Seldinger technique), and access was permitted via femoral vessels. A 16- to 18-Fr arterial cannula and a 21- to 24-Fr venous cannula were used depending on the patient’s blood vessel diameter. Distal perfusion cannulas (DPC) were routinely placed because of the poor distal artery flow after peripherally placed cannulas. 

After ECMO catheterization, cranial Computed Tomography perfusion (CTP) imaging was routinely performed within 6 h; cerebral imaging with hemorrhage, hypoperfusion, or ischemic penumbra was excluded. During ECMO operation, daily GCS scores, long-range EEG monitoring (24 h), and cranial CT examination were routinely performed. The cranial MR examination was performed as soon as possible while ensuring patient safety to reduce the neurological damage that occurs after ECMO. Patients with neurological damage after ECMO removal were excluded. Patients who could not cooperate with MR imaging one week after ECMO removal were excluded. The inclusion criteria were as follows: (1) Age > 18 years. (2) No previous history of neurological diseases. (3) VA ECMO support due to cardiogenic shock. (4) Survived after weaning and can tolerate MRI examination. The exclusion criteria were as follows: (1) Unable to remove ECMO. (2) Inability to cooperate or complete MR imaging. (3) Increased serum levels of NSE were accompanied by elevated free hemoglobin. (4) The presence of tumor or intracranial vascular malformation on MR imaging. 

Currently, the most widely used Coma Index, a medical indicator of the degree of coma in patients, is the Glasgow Coma Index (GCS—Glasgow Coma Scale). This index was published by Graham Teasdale and Bryan J. Jennett, two professors of neurosurgery at the University of Glasgow, in 1974 [9]. The factors affecting GCS score are as follows: (1) GCS evaluation is not appropriate for the use of sedatives. It is estimated that GCS evaluation should be reevaluated when there is no drug effect. (2) Patients with epileptic brain disorders often have seizures, especially when status epilepticus is still in a coma during the interictal period. Attention should be paid to distinguish it from a coma caused by the primary disease. There are several other points that cannot be ignored in the evaluation of consciousness. (a) When judging, the best response should be used to calculate the score. (b) Peripheral pain stimulation should be used when the pain stimulation is scored, and the pain stimulation should be from light to heavy to avoid unnecessary pain. Stimulation can be repeated, but not for too long. More importantly, when assessing an individual patient, it is best described in terms of the three components of the coma scale. It is not just a total score, because patients with the same GCS score may have different disease states. During a single examination, some patients gave variable responses that generally improved as the patient became more excited. The left and right limbs may also respond differently. Any difference between limb and other limb responses may indicate focal brain damage, for which the worst response should be noted, but to assess the extent of altered consciousness, we record the best response of the best limb. Confounding factors made one or more components of the Glasgow Coma Scale untenable. 

Glasgow Coma Scale (GCS) assessment was performed from the first day of ECMO support until the patient was successfully taken off ECMO; the GCS was also assessed after the patient was taken off ECMO for five and a half lives to eliminate the interference of drug sedation and mild hypothermia. For all included patients, we discontinued the sedative drug for at least five and a half lives after the end of brain protection. The bi-spectral (BIS) index indicated that it was greater than 90, and the blood drug concentration was determined. If the blood drug concentration reached the standard, a GCS evaluation was performed again; the evaluation was performed several times a day. For our study, we took the highest value of the day and recorded it. We recorded the worst GCS score and GCS motor response in our statistical analysis.

Near-infrared spectroscopy (NIRS) measurement of local brain tissue oxygen saturation (StO2) enables continuous non-invasive assessment of oxygen delivery (DO 2) and oxygen consumption (VO 2) in frontal brain regions. StO2 Reflects the mixed saturation of the measured cerebral microcirculation, consisting of arterial (A), venous (V), and capillary signals, where the A: V signal ratio fluctuates according to DO 2 and VO 2 in brain tissue. The NIRS technique relies on two or four light attenuation changes at different wavelengths and transforms them into changes in the concentration of three chromophores (oxyhemoglobin, deoxyhemoglobin, and cytochrome oxidase). In our included study group, NIRS was routinely used to measure brain tissue oxygen for each VA ECMO patient in our center. If the brain oxygen index was continuously less than 60, it was determined as brain tissue hypoxia.

### 2.1. Serum Samples and Assays

Blood samples were drawn during the ECMO operation. Blood was centrifuged at 2643× *g* for 10 min. Hemolytic samples (hemoglobin concentration above 47 mg/dL) were excluded. Serum neuron-specific enolase (NSE) assays were performed using immunofluorimetric assays and electrochemiluminometric sandwich immunoassays (Kryptor^®^, Brahms, Brandenburg, Germany, and Modular^®^ E170, Roche Diagnostics, Basel, Switzerland). Detection of NSE serum levels started with ECMO operation and continued until the evacuation of ECMO. The maximum value of the NSE level was included in our statistical analysis.

The control group was individuals who attended our hospital for annual health check-ups and had no history of neurologic or severe cardiovascular diseases.

### 2.2. Calculation of MOCA

The Montreal Cognitive Assessment Scale (MoCA Scale) is an assessment tool for the rapid screening of cognitive dysfunction. It takes about 15 min and includes 11 inspection items in 8 cognitive domains, including attention and concentration, executive function, memory, language, visual structure skills, abstract thinking, computation, and directional force. The total score was 30; 26 was considered normal, with lower scores indicating greater cognitive impairment. The following requirements should be paid attention to during the evaluation: (1) The MoCA scale out of 30: 26 points are normal, mild cognitive dysfunction (MCI) scores are about 19–25, and AD scores are lower. (2) If the subject has received 12 years of education (high school level), one point can be added to the result, but the total score should not exceed 30 points. If the patient is illiterate or poorly educated, the base scale (basic test) can be selected. (3) The speed of words or numbers read is one per second. (4) If the patient is unable to write due to a disability (such as hemiplegia), the full score is 25 points, and the final score is converted to a 30-point system. All subjects included in this study were tested for MOCA by a professionally qualified neurologist in our hospital.

### 2.3. Magnetic Resonance Imaging Protocol

Whole-brain MRI data were acquired at the Center for Brain Imaging Science and Technology, First Affiliated Hospital of Zhejiang University School on a Siemens. 

MAGNETOM Prisma 3T scanner (Siemens, Erlangen, Germany). All participants were placed in the machine with foam padding around the head to reduce motion; they were asked to keep still with their eyes closed during imaging.

Echo-planar imaging sequence was used to acquire the functional images with the following features: 60 axial slices, thickness/gap = 2.0/0 mm, in-plane resolution = 64 × 64, repetition time (TR) = 2000 ms, echo time (TE) = 34 ms, flip angle = 62° and field of view (FOV) = 220 × 220 mm^2^. Anatomical T1-weighted whole-brain magnetization-prepared rapid gradient echo images were obtained using the following: 160 sagittal slices, slice thickness/gap = 1.2/0 mm, in-plane resolution = 512 × 512, TR = 5000 ms, TE = 2.9 ms, inversion time (TI) = 700 ms, flip angle = 4° and FOV = 256 × 256 mm^2^.

### 2.4. Processing of MRI Data

SPM8 (http://www.fil.ion.ucl.ac.uk/spm (accessed on 28 November 2022)) was used to implement pre-processing of all fMRI data while data processing was conducted using the Data Processing Assistant for Resting-State fMRI (http://www.restfrmi.net (accessed on 28 November 2022)). The initial 10 volumes of the functional images were discarded to remove initial transient effects and to allow the participant to adjust to the scanner noise before pre-processing. The rest of the fMRI images were acquired with slice timing for the acquisition delay between slices and correction of head motion. All participants who were under imaging had less than 1.5 mm maximum displacement in x, y, or z and 1.5° angular motion during imaging. Spatial normalization and resampling to 3 mm voxels were used to acquire realigned images, while a Gaussian filter (6 mm FWHM) was used to spatially smoothen the images. Smoothened images were filtered using a typical temporal bandpass (0.01–0.08 Hz) to reduce low-frequency drift, as well as physiological high-frequency respiratory and cardiac noise. Linear trends were removed within each time series. Lastly, spurious variances from several sources were removed by linear regression, including six head motion parameters, along with average signals from cerebrospinal fluid and white matter.

### 2.5. Calculation of DC

Voxel-based whole-brain correlation analysis of pre-processed fMRI was performed to calculate voxel-wise DC, as previously described [10]. Pearson’s correlation coefficients (r) were determined between all pairs of brain voxels in the gray matter mask. We then converted the Pearson’s correlation data to normally distributed Fisher’s Z-scores and constructed the whole-brain functional network by thresholding each correlation at r > 0.25, as previously reported [11]. The DC for a given voxel was calculated as the sum of the significant connections at the individual level. Voxel-wise DC values were also converted into a Z-score map using the Fisher-Z transformation to improve normality. Positive correlations were considered in the DC calculation due to the uncertainty of interpretation and detrimental effects on test–retest reliability.

To assess the DC difference between ECMO survivors and controls, a two-sample *t*-test was performed using REST. AlphaSim, a program based on Monte Carlo simulation and implemented in AFNI (http://afni.nimh.nih.gov (accessed on 28 November 2022)), was used for multiple comparison corrections. Monte Carlo simulations determine the random distribution of cluster size for a given per voxel threshold [12]. Statistical difference was defined as *p* < 0.05 and cluster size >198 voxels, corresponding to a corrected *p* < 0.05. The correction was confined within the gray matter mask and was determined by Monte Carlo simulations [12].

All participants underwent a Montreal Cognitive Assessment, MoCA, to screen for cognitive decline [13]. MoCA was performed 24 h after each participant’s MR imaging. Participants were required to recall words from MoCA again after a 20 min delay as long-delay recall (DR).

### 2.6. Statistical Analysis

To examine the differences between patients and controls, the chi-square test was used for categorical variables and ANOVA for continuous and normally distributed variables. A linear regression model was used to assess the correlation between MRI variables and clinical insinuations. All regression analyses were adjusted for age, gender, hypertension, diabetes, and education (for cognitive measures). Data analysis and plotting were performed in R version 4.0.3 (R Core Team 2020, Rahway, NJ, USA). *p* values less than 0.05 were considered statistically significant.

## 3. Results

Figure 1 shows the schematic overview and flow chart of our enrolled participants. Table 1 shows the characteristics of the study participants used in our data analysis. Twenty-seven ECMO survivors and twenty-eight controls were included. ECMO survivors had lower MoCA scores and a higher burden of hypertension and diabetes. Importantly, the ECMO running time was 5.1 days. Out of the 27 ECMO survivors, 13 had full flow support while 14 had non-full flow support. The average running time for full-flow support was 4.7 days, while the average running time for non-full-flow support was 5.5 days.

Table 2 and Figure 2c show the differences in DC values between ECMO survivors and controls. DC values in the right insula and right precuneus gyrus were lower in ECMO survivors and higher in the right medial superior frontal gyrus when compared to controls (all *p* < 0.001). 

Figure 2d displays the association between DC values and clinical insinuations in ECMO survivors. Decreased connectivity in the right insular and right precuneus gyrus correlated with total MoCA scores, delayed recollection, and calculation (all *p* < 0.05). Increased serum NSE levels, GCS score, and GCS m correlated with decreased connectivity in the right insular and right precuneus gyrus and increased connectivity in the right medial superior frontal gyrus (all *p* < 0.05). 

## 4. Discussion

VA ECMO is an important means of treating critically ill patients. Its neurological complications are directly related to its poor prognosis. Functional MRI is an objective and good index with which to evaluate brain function. The purpose of this study was to investigate the changes in resting-state fMRI in ECMO survivors. Large clinical studies related to ECMO and fMRI are currently limited. To our knowledge, this is the first study to investigate the cerebral connectivity changes in ECMO survivors and explore the association with their clinical insinuation. Our study showed that ECMO survivors showed reduced connectivity in the right insula and right precuneus gyrus and increased connectivity in the right medial superior frontal gyrus when compared to controls. We also showed these cerebral changes in ECMO correlated with their cognitive performance, neurological deficits, and operation.

Previous autopsy reports [14,15,16] showed various forms of neuropathology such as hemorrhages, infarctions, and hypoxic–ischemic injury were seen in autopsies of adults who received ECMO before death. While it remains unknown whether ECMO causes these injuries, numerous mechanisms make this a possibility. However, in vivo reports on ECMO survivors are limited. 

Functional MRI is an objective and good index with which to evaluate brain function. At present, the research on the prognosis of ECMO neurological function is mostly limited to EEG, some blood biochemical indicators, and head CT examination, and none of these involve functional magnetic resonance. Brain injury due to hypoxia is a well-known complication of ECMO support [17,18,19,20]; reduced cerebral blood flow patterns can result in neurological disorders in survivors of ECMO [20,21,22,23]. Our results showed that after ECMO, survivors had decreased connectivity in the right insula and right precuneus gyrus and increased connectivity in the right medial superior frontal gyrus compared to controls. Cerebral hypoxia has always been a serious complication of VA ECMO, which is directly associated with a poor prognosis. Current monitoring methods include brain tissue oxygen, real-time monitoring of TCCD, and maintaining appropriate intracranial perfusion through ECMO and drugs. The main pathological basis is the acute or chronic decline of cardiac systolic or diastolic function, which leads to a decrease in cardiac output and cannot meet its own metabolic needs. This may have resulted in changes in neural connectivity due to the reduced cardiac output, ultimately resulting in cerebral hypoxia. We suggest that decreased connectivity may reflect neural lesions in these areas, while increased connectivity may be a compensatory effect involving functional reorganization of the damaged brain. Taken together, our results showed both functional impairment and adaptation were observed in survivors of ECMO, suggesting that neural connectivity changes may provide insights into the mechanisms associated with cerebral changes. 

Given the many risk factors for cognitive impairment during and after ECMO treatment, assessment of cognition is of vital importance. It is suggested that ECMO survivors often experience cognitive dysfunction [24,25]. Compared to controls, ECMO survivors showed reduced cognitive scores, indicating cognitive decline; this is in line with previous reports [20,26,27]. We noted decreased connectivity in the right insula and right precuneus correlated with executive function, delayed recall, and total MoCA score in survivors after ECMO after adjusting for confounding factors. Since these cerebral areas are associated with cognition in an individual, the association between the cognitive measures and reduced connectivity suggests that neural impairment may affect their cognition. 

Neuron-specific enolase (NSE) is an enzyme unique to neurons and neuroendocrine cells. As a sensitive index used to evaluate the severity of nerve cell injury and judge the prognosis, it is widely used in various nerve injuries and is a commonly used clinical and experimental sensitive index that evaluates the severity of the craniocerebral injury and judges the disease prognosis. In general, NSE is almost absent in serum and cerebrospinal fluid. After cranial nerve injury, part of the neurons is necrotic and disintegrated, and the integrity of the cell membrane of cranial nerve tissue is damaged, forcing NSE in nerve cells to diffuse into the cerebrospinal fluid and intercellular space. Due to the destruction of the blood–brain barrier caused by injury, its integrity is destroyed, or its permeability is enhanced, and its natural barrier function is weakened, resulting in the release of some protein components through the blood–brain barrier into the blood and cerebrospinal fluid. The levels of NSE in blood and cerebrospinal fluid increased with the severity of brain injury, the number of dead and disintegrated neurons, and the severity of blood–brain barrier damage. This mechanism has become a theoretical basis for detecting the changes of NSE to judge the degree of nerve damage after brain nerve injury. Therefore, it is considered that NSE is a specific and sensitive index to judge the degree of brain injury and the prognosis of disease.

NSE is primarily released by neurons in the setting of injury. A recent study demonstrated that high serum levels of NSE were related to neurologic complications [28,29]. Our study showed decreased connectivity in the right insula and right precuneus gyrus and increased connectivity in the right superior frontal gyrus correlated with increased NSE serum levels. It is plausible to suggest that the association between high NSE levels and cerebral neuronal connectivity changes in ECMO survivors may reflect the neuronal-impaired events. Monitoring of NSE levels along with neuroimaging parameters may help identify the degree of nerve cell damage that occurs during ECMO operation.

The GCS is a very simple method for neurological monitoring and is suggested to provide information to identify patients with favorable neurological outcomes. Our results showed cerebral connectivity changes correlated with total GCS score and motor responses in GCS in ECMO survivors. The GCS score is a comprehensive assessment of brain function, and we speculate that changes in neural network connectivity may be one of the factors affecting the GCS score in ECMO survivors.

We would like to acknowledge some limitations of our study. First, our study is an observational cross-section design; thus, our results cannot confirm the causal relationship between ECMO provision and the downstream cognitive decline and neurological deficit. We conducted this study on ECMO survivors, leaving open the possibility of selection bias due to the omission of a non-ECMO control group. Future studies examining the cerebral changes in ECMO survivors should include a control group matched by the severity of illness and demographics. As with most neuroimaging tools, patient cooperation is an obligation. Movement from participants can reduce the quality of the image, which may affect the data. Neurofunctional imaging examination is the objective basis for evaluating brain function. At present, there are almost no clinical studies on brain functional imaging of ECMO survivors. The absence of a large number of reference experiments inevitably leads to subjective deviation in our understanding of the results. More large, longitudinal clinical studies may help inform our understanding of the pathophysiology of neurofunctional changes in ECMO survivors. Regarding the subject of fMRI changes in ECMO survivors, a larger sample from multicenter, longitudinal clinical studies is needed. Degree centrality (degree centrality, DC) is a voxel-based level analysis method that analyzes the immediate functional connections of a voxel and other voxels in the brain, reflecting the functional importance of this voxel in the brain network. Moreover, many studies show higher reliability and repeatability of DC than other indicators. However, a description of gray matter and white matter volume changes in brain structure is lacking in this study. Another limitation of our study was that PaO2 was not analyzed in our study; future studies should include PaO2 values. The effect of cardiotropic drugs and vasoactive drugs on the prognosis of neurological function is indeed a noteworthy problem. This study is a single-center, small-sample study. All the included patients were in cardiogenic shock, and the vasoactive drugs used under ECMO support were mainly due to decreased vascular tone caused by sedatives. If drugs are grouped according to different drug groups, there will be a large difference in the number of subgroups, affecting the statistical results.

## 5. Conclusions

In conclusion, we showed that both functional impairment and adaptation were observed in survivors of ECMO, suggesting that neural connectivity changes may provide insights into the mechanisms that may potentially link ECMO survivors to neurological and cognitive disorders. Further research is necessary to elucidate the potential mechanisms underlying these relationships. 

## Figures and Tables

**Figure 1 brainsci-13-00630-f001:**
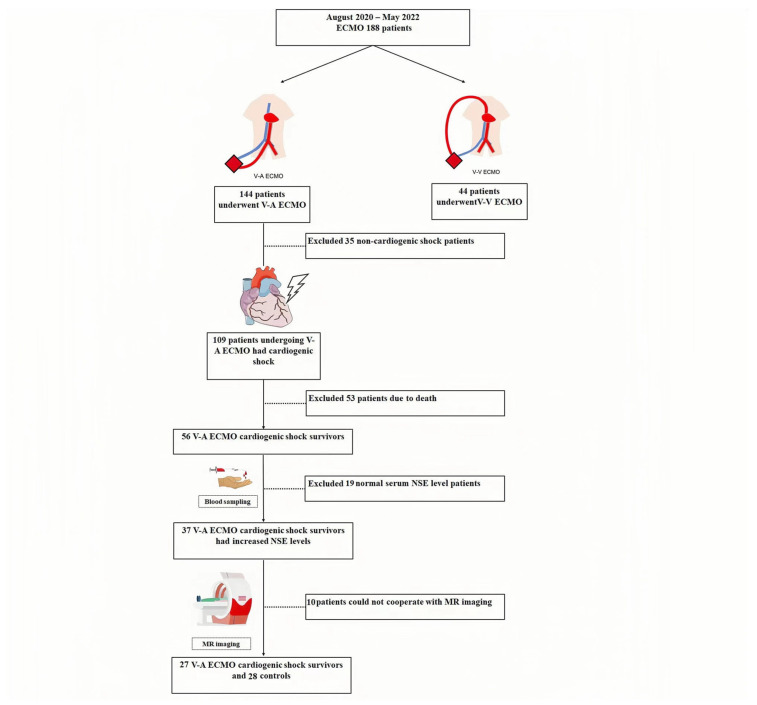
Schematic flow chart of our study participants.

**Figure 2 brainsci-13-00630-f002:**
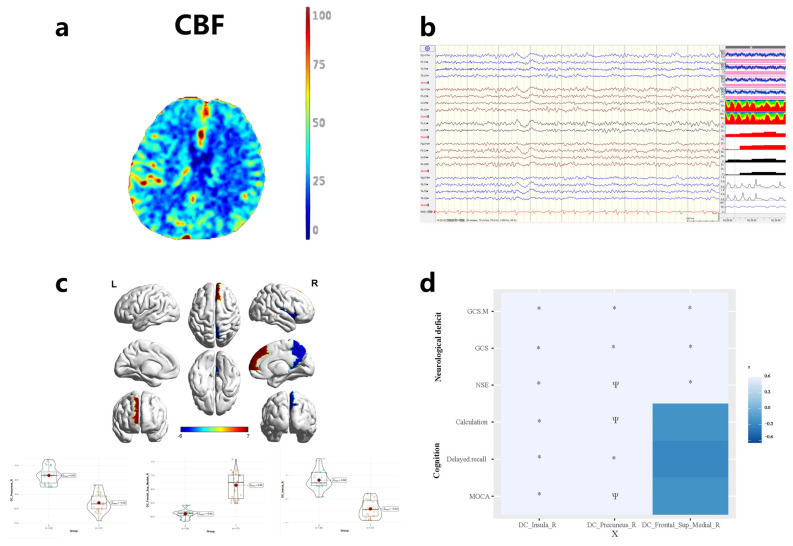
(**a**) Cerebral blood flow image of Cranial Computed Tomography perfusion after ECMO support. (**b**) Long-range EEG monitoring during the ECMO operation. (**c**) Comparison of brain regions in ECMO survivors compared to controls. insula_R: right insula; precuneus_R: right precuneus gyrus; Frontal_sup_medial_R: right superior medial frontal gyrus. (**d**) Association between DC changes in ECMO survivors and their clinical measures. insula_R: right insula; precuneus_R: right precuneus gyrus; Frontal_sup_medial_R: right superior medial frontal gyrus; MoCA: Montreal Cognitive Assessment; GCS: Glasgow Coma Scale. * stands for P < 0.05 and Ψ stands for P < 0.001.

**Table 1 brainsci-13-00630-t001:** Demographics of our study participants.

	ECMO Survivors	Controls	*p*-Value
Number	27	28	
Age, years	43.04 ± 9.55	42.89 ± 9.05	0.954
Gender, males	11	13	0.885
Hypertension, *n*	5	4	
Diabetes, *n*	3	4	
Dyslipidemia, *n*	3	2	
Smokers, *n*	3	3	
Drinkers, *n*	3	2	
Education, years	9 (8–14)	9 (8–14)	0.471
MoCA score	25 (24–26)	29 (29–30)	<0.001
Serum NSE	52 (41–67)		
GCS score	7 (6–8)		

ECMO: Extracorporeal Membrane Oxygenation; MoCA: Montreal Cognitive Assessment; NSE: neuron-specific enolase; GCS: Glasgow Coma Scale.

**Table 2 brainsci-13-00630-t002:** Comparison of cerebral DC regions in ECMO survivors compared to controls.

Brain Regions	Voxels	BA	MNI Coordinates			*p* Value
			X	Y	Z	
Insula_R	39	48	39	−3	6	<0.001
Precuneus_R	43		9	−63	33	<0.001
Frontal_Sup_Medial_R	64	32	12	33	39	<0.001

insula_R: right insula; precuneus_R: right precuneus gyrus; Frontal_sup_medial_R: right superior medial frontal gyrus.

## Data Availability

The data supporting the conclusions of this article will be made available by the authors upon reasonable request.

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
