# Peer review of "Cerebral Neural Changes in Venous–Arterial Extracorporeal Membrane Oxygenation Survivors"

_brainsci, 2023, doi:10.3390/brainsci13040630_

Round 1

Reviewer 1 Report

the authors presented a fine work on cerebral changes in patients who underwent ECMO. I have some few pointers that need to be addressed:

1. why compare the cerebral changes to controls when we know that the changes will be severe in the disease group

2. why choose MoCA when there are many neuropsychological tools?

3. was GCS done during ECMO or after... did the GCS change after ECMO

Author Response

  1. why compare the cerebral changes to controls when we know that the changes will be severe in the disease group

Reply: these controls were not without any neurological disorders. However, to find the true effect of brain changes in our ECMO survivors we wanted to see how these changes affected the brain. This was done to pinpoint the changes in the brain in ECMO survivors to highlight its neurological damage.

  1. why choose MoCA when there are many neuropsychological tools?

Reply: MoCA has been suggested to be very sensitive to neuropsychological changes especially in the brain. Thus, we chose MoCA

  1. was GCS done during ECMO or after... did the GCS change after ECMO

Reply: GCS was done during ECMO

Reviewer 2 Report

This reviewer has read with interest this manuscript that investigates the impact of V-A ECMO on cognitive and neurological measurements. This approach is interesting as very little is known about the potential neurological changes after V-A ECMO.

Comments

What is unclear to me is how the authors defined GCS during ECMO? Was sedation completely stopped? Please comment and makes this more clear in the methods section.

What was the base pathology of the cardiac failure in the ECMO patients? Provide more information in a table with respect to the ECMO patients.

What was the amount of vasopressors given during ECMO support? Was any correlation found between the amount of vasopressor use and the cognitive and neurological measurements?

What was the average duration of ECMO? Was any correlation found between duration and cognitive and neurological measurements?

What do the authors mean by: "After ECMO removal, the cranial MR examination was performed as soon as possible while ensuring patient safety to reduce neurological damage that occurs after ECMO withdrawal." Which neurological damage do they allude to?

As this were all V-A ECMO patients with peripheral cannulation, how were they monitored to avoid brain hypoxia during support? Which measures were taken by the authors in case of Harlequin syndrome? V-AV ECMO? This information is important as a decreased connectivity was found in the right insula.

Were PaO2 values during ECMO used in the analysis? How many patients did encounter hypoxia during ECMO support? What was the hemoglobin concentration during ECMO support? Were any correlations found between DO2 and the neurological measurements?

During V-A ECMO often partial support is used. How many patients were on full support and for how long and how many patients were on partial support and for how long. What was the average time necessary to wean the patients?

It is good to notice that the authors added the MoCA to the analysis. This reviewer strongly believes that neurocognitive tests reveal additional information.  However, I am not sure that this gives reliable results immediately after ECMO removal due to the fact that patients still may have some residual impact of sedation and other drugs. Why was this test not repeated for example before leaving the hospital? Please comment.

Author Response

Q1 What is unclear to me is how the authors defined GCS during ECMO? Was sedation completely stopped? Please comment and makes this more clear in the methods section.

Reply: At present, it is difficult to completely rule out the influence of sedative drugs on the GCS score, especially for ICU patients. In all included patients, we discontinued the sedative drug for at least 5 days after the end of brain protection, and the bi-spectral (BIS) index indicated that it was greater than 90 and blood drug concentration was detected. If the blood drug concentration reached the standard, GCS evaluation was performed, and the evaluation was performed several times a day. For our study, we took the highest value of the day and recorded it. (Methods Section)

Q2 What was the base pathology of the cardiac failure in the ECMO patients? Provide more information in a table with respect to the ECMO patients.

Reply: The main pathological basis is the acute or chronic decline of cardiac systolic or diastolic function, which leads to a decrease in cardiac output and cannot meet its own metabolic needs. Due to the small sample size of the included patients, no etiological grouping was performed, which is also the shortcoming of this paper. (Discussion section)

A3 What was the amount of vasopressors given during ECMO support? Was any correlation found between the amount of vasopressor use and the cognitive and neurological measurements?

Reply: The patients included in this study had complex and critical conditions; thus the patients used in our had different therapy routine with different dosage. This caused a difficulty in our statistical analysis, and we did not find a good statistical correlation.

Q4 What was the average duration of ECMO? Was any correlation found between duration and cognitive and neurological measurements?

Reply: The average ECMO running time was 5.1 days. No correlation was found in the statistical results. (Methods Section)

Q5 What do the authors mean by: "After ECMO removal, the cranial MR examination was performed as soon as possible while ensuring patient safety to reduce neurological damage that occurs after ECMO withdrawal." Which neurological damage do they allude to?

Reply: We have revised that portion. The condition of our included patients is complex, and the research group is mainly designed to minimize the interpretation of impaired neurological function caused by non-ECMO factors.

Q6 As this were all V-A ECMO patients with peripheral cannulation, how were they monitored to avoid brain hypoxia during support? Which measures were taken by the authors in case of Harlequin syndrome? V-AV ECMO? This information is important as a decreased connectivity was found in the right insula.

Reply: Cerebral hypoxia has always been a serious complication of VA ECMO, which is directly associated with a poor prognosis. Current monitoring methods include brain tissue oxygen, real-time monitoring of TCCD, and maintaining appropriate intracranial perfusion through ECMO and drugs. However, this cannot completely avoid the hypoxia of the brain during VA ECMO maintenance and other complications caused by ECMO, which is also the purpose of this study. If conventional ventilator adjustment and pulmonary physiotherapy will not improve the oxygenation. The VAV is a coping approach. (Discussion portion)

Q7 Were PaO2 values during ECMO used in the analysis? How many patients did encounter hypoxia during ECMO support? What was the hemoglobin concentration during ECMO support? Were any correlations found between DO2 and the neurological measurements?

Reply: The PAO 2 was not analyzed in this study. Due to the complex and critical condition of the patients included in this study, the ECMO maintenance time was long, and 15 / 27 patients with brain hypoxia were found, but subsequent magnetic resonance confirmed that all included patients had brain tissue damage. The mean hemoglobin concentration was 6.5 g/dl. We did not do an analysis of DO 2 and neurological prognosis, but each of our patients did internal jugular venous blood gas to determine whether the oxygen supply and consumption of the brain matched. However, the results did not find a good correlation. The reason may be that the internal jugular venous blood gas oxygen saturation may also be maintained at normal levels in the presence of severe cerebral hypoxia.

Q8 During V-A ECMO often partial support is used. How many patients were on full support and for how long and how many patients were on partial support and for how long. What was the average time necessary to wean the patients?

Reply: 13 achieved full flow support and 14 had non-full flow support. The average running time of full flow support is 4.7 days. The average running time of the non-full flow rate was 5.5 days. (Results Section)

Q9It is good to notice that the authors added the MoCA to the analysis. This reviewer strongly believes that neurocognitive tests reveal additional information.  However, I am not sure that this gives reliable results immediately after ECMO removal due to the fact that patients still may have some residual impact of sedation and other drugs. Why was this test not repeated for example before leaving the hospital? Please comment.

Reply: The influence of sedative drugs on MOCA and GCS scores is of a great concern. All of our subjects have discontinued sedative and analgesic drugs for more than 5 days, the BIS index is greater than 90, and the blood concentration has reached the standard, excluding drug interference to the best possible. Repeated evaluation of MOCA score in the short term cannot effectively evaluate the real cognitive level of patients. Therefore, the repeat test was not performed.

Reviewer 3 Report

1.      The paper presentation is not clear. What is the objective of the conducted research? What is the novelty related scientific literature ?It seems to me that the authors have conducted some experiments on a dataset without really proposing a "system"

Identify the main findings and justify the novelty and contribution of the work

Please, clearly explain how your solution advances existing approaches

2.      It would be better if the authors added two separate sections: the introduction with the main contributions and the related works that present just the methods for the last five years. Please show the limitations in the previous works and what is the novelity in your proposed work.

3.      The related work section should be elaborated by including more papers and clear explanation of existing methodologies.

4.      The descriptions given in this proposed scheme are not sufficient that this manuscript only adopted a variety of existing methods to complete the experiment where there are no strong hypothesis and methodical theoretical arguments. Therefore, the reviewer considers that this paper needs more works

5.      Does the proposed method have some shortcomings? In fact, shortcomings don’t reduce the availability of the proposed method. By contrast, it is a very suitable way to help readers to understand the proposed method comprehensively in my opinion.

6.      In the Conclusion section, please explain more about future works

Author Response

1.The paper presentation is not clear. What is the objective of the conducted research? What is the novelty related scientific literature? It seems to me that the authors have conducted some experiments on a dataset without really proposing a "system". Identify the main findings and justify the novelty and contribution of the work. Please, clearly explain how your solution advances existing approaches.

Reply: VA ECMO is an important means of treating critically ill patients. Its neurological complications are directly related to its poor prognosis. Functional MRI is an objective and good index to evaluate brain function. The purpose of this study was to investigate the changes in resting-state fMRI in ECMO patients. Large clinical studies related to ECMO and fMRI are currently unavailable. This study is the first of its kind, but it does lack references from other large-scale studies, and the sample size is small, but the data have significant statistical differences. We believe that it still has certain clinical value and provides a preliminary theoretical basis for later larger clinical studies. (Discussion Section)

  1. It would be better if the authors added two separate sections: the introduction with the main contributions and the related works that present just the methods for the last five years. Please show the limitations in the previous works and what is the novelty in your proposed work.

Reply: Functional MRI is an objective and good index to evaluate brain function. At present, the research on the prognosis of ECMO neurological function is mostly limited to EEG, some blood biochemical indicators and head CT examination, and none of them involves functional magnetic resonance. This study is the first of its kind, using functional magnetic resonance for the first time to study the neurological changes of ECMO patients, and to provide a preliminary theoretical basis for future larger studies. (Discussion section)

  1. The related work section should be elaborated by including more papers and clear explanation of existing methodologies.

Reply: To the best of our knowledge, this study is the first thus, other clinical research support and reference are currently lacking.

 4.The descriptions given in this proposed scheme are not sufficient that this manuscript only adopted a variety of existing methods to complete the experiment where there are no strong hypothesis and methodical theoretical arguments. Therefore, the reviewer considers that this paper needs more works。

Reply: This study is the first of its kind, and there is currently a lack of valid references. Therefore, our team uses the classic magnetic resonance analysis method to analyze brain function. We have already started projects with other, more novel methods for analysis for follow-ups.

  1. Does the proposed method have some shortcomings? In fact, shortcomings don’t reduce the availability of the proposed method. By contrast, it is a very suitable way to help readers to understand the proposed method comprehensively in my opinion.

Reply: Degree centrality (degree centrality, DC) is a voxel-based level analysis method that analyzes the immediate functional connections of a voxel and other voxels in the brain, reflecting the functional importance of this voxel in the brain network. Moreover, many studies show higher reliability and repeatability of DC than other indicators. However, a description of gray matter and white matter volume changes in brain structure is lacking in this study.

  1. In the Conclusion section, please explain more about future works

Reply: Regarding the subject of fMRI changes in ECMO patients, larger sample from multicenter with longitudinal study design clinical studies are needed. (Discussion Section)

Round 2

Reviewer 2 Report

Thank you for the clarifications given, and the adaptations made to the original manuscript.

What is still not clear to me is that the authors claim on one hand that GCS scale was done from the first day of ECMO but on the other hand state that: "In all included patients, we discontinued the sedative drug for at least 5 days after the end of brain protection". Please clarify.

Please clarify what your center considers "full flow", especially taken in consideration that the average hemoglobin concentration was relatively low during ECMO.

Several studies show a relationship between the VIS score and neurological outcome. Please be more precise about inotropic support during ECMO.

The authors mention that 15 / 27 patients had brain hypoxia. How did they monitor for brain hypoxia during ECMO. I understood they measured jugular bulb saturation but what else was used routinely? Please also clarify the algorithm used when brain hypoxia was noted.

Author Response

Thank you for the clarifications given, and the adaptations made to the original manuscript.

What is still not clear to me is that the authors claim on one hand that GCS scale was done from the first day of ECMO but on the other hand state that: "In all included patients, we discontinued the sedative drug for at least 5 days after the end of brain protection". Please clarify.

Reply: What we meant was "stop the drug for more than five half-lives, not five days". It was also stated in the paper that all patients are evaluated daily for GCS, which is our clinical practice; The data included in the final analysis of each patient in the paper were required to exclude sedatives. I think it is the way of our description that caused you difficulties.

Please clarify what your center considers "full flow", especially taken in consideration that the average hemoglobin concentration was relatively low during ECMO.

Reply: "full flow" is defined as the heart does not have any output function, the heart has no systolic function as indicated by the ultrasound, the arterial blood pressure indicates no pulse pressure, and the cardiac output needs to be fully supported by the ECMO. Although the hemoglobin concentration of patients is low, we will actively inject blood products for ECMO patients to maintain the hemoglobin level as high as possible. Meanwhile, we will calculate DO2 and VO2 to ensure that DO2/VO2 is greater than 3/1.

Several studies show a relationship between the VIS score and neurological outcome. Please be more precise about inotropic support during ECMO.

Reply: The effect of cardiotropic drugs and vasoactive drugs on the prognosis of neurological function is indeed a noteworthy problem. Because this study is a single-center, small-sample study. All the included patients were in cardiogenic shock, and the vasoactive drugs used under ECMO support were mainly due to decreased vascular tone caused by sedatives. If drugs are grouped according to different drug groups, there will be a large difference in the number of subgroups, affecting the statistical results. This has been stated in limitations

The authors mention that 15 / 27 patients had brain hypoxia. How did they monitor for brain hypoxia during ECMO. I understood they measured jugular bulb saturation but what else was used routinely? Please also clarify the algorithm used when brain hypoxia was noted.

Reply: Near-infrared spectroscopy (NIRS) is routinely used to measure brain tissue oxygen for each VA ECMO patient in our center. If the brain oxygen index is continuously less than 60, it is determined as brain tissue hypoxia. However, the effective area measured by NIRS is small and lacks effective sensitivity to hypoxia in brain tissue. Subsequent MRI showed that the number of patients with impaired neurological function was greater than the results found by NIRS. This has been added to our revised manuscript (Methods Section)

Reviewer 3 Report

The author performed all the required modification

Author Response

Thank you for your useful suggestions